# Predicting the Invasiveness of Pulmonary Adenocarcinomas in Pure Ground-Glass Nodules Using the Nodule Diameter: A Systematic Review, Meta-Analysis, and Validation in an Independent Cohort

**DOI:** 10.3390/diagnostics14020147

**Published:** 2024-01-08

**Authors:** Jieke Liu, Xi Yang, Yong Li, Hao Xu, Changjiu He, Peng Zhou, Haomiao Qing

**Affiliations:** Department of Radiology, Sichuan Clinical Research Center for Cancer, Sichuan Cancer Hospital and Institute, Sichuan Cancer Center, Affiliated Cancer Hospital of University of Electronic Science and Technology of China, Chengdu 610041, China; liu.jieke@uestc.edu.cn (J.L.); m17726362884@163.com (X.Y.); muzi5969@foxmail.com (Y.L.); xu378531592@foxmail.com (H.X.); hechangjiou@sina.com (C.H.)

**Keywords:** pure ground-glass nodule, invasive adenocarcinoma, diameter, computed tomography, meta-analysis

## Abstract

The nodule diameter was commonly used to predict the invasiveness of pulmonary adenocarcinomas in pure ground-glass nodules (pGGNs). However, the diagnostic performance and optimal cut-off values were inconsistent. We conducted a meta-analysis to evaluate the diagnostic performance of the nodule diameter for predicting the invasiveness of pulmonary adenocarcinomas in pGGNs and validated the cut-off value of the diameter in an independent cohort. Relevant studies were searched through PubMed, MEDLINE, Embase, and the Cochrane Library, from inception until December 2022. The inclusion criteria comprised studies that evaluated the diagnostic accuracy of the nodule diameter to differentiate invasive adenocarcinomas (IAs) from non-invasive adenocarcinomas (non-IAs) in pGGNs. A bivariate mixed-effects regression model was used to obtain the diagnostic performance. Meta-regression analysis was performed to explore the heterogeneity. An independent sample of 220 pGGNs (82 IAs and 128 non-IAs) was enrolled as the validation cohort to evaluate the performance of the cut-off values. This meta-analysis finally included 16 studies and 2564 pGGNs (761 IAs and 1803 non-IAs). The pooled area under the curve, the sensitivity, and the specificity were 0.85 (95% confidence interval (CI), 0.82–0.88), 0.82 (95% CI, 0.78–0.86), and 0.73 (95% CI, 0.67–0.78). The diagnostic performance was affected by the measure of the diameter, the reconstruction matrix, and patient selection bias. Using the prespecified cut-off value of 10.4 mm for the mean diameter and 13.2 mm for the maximal diameter, the mean diameter showed higher sensitivity than the maximal diameter in the validation cohort (0.85 vs. 0.72, *p* < 0.01), while there was no significant difference in specificity (0.83 vs. 0.86, *p* = 0.13). The nodule diameter had adequate diagnostic performance in differentiating IAs from non-IAs in pGGNs and could be replicated in a validation cohort. The mean diameter with a cut-off value of 10.4 mm was recommended.

## 1. Introduction

Lung cancer remains the most common cause of cancer-related mortality worldwide [1,2]. The detection of subsolid nodules, including pure ground-glass nodules (pGGNs) and mixed ground-glass nodules (mGGNs)/part-solid nodules (PSNs), has expanded enormously with the popularization of low-dose computed tomography (LDCT) in lung cancer screening, particularly in Eastern Asians [3,4,5,6,7]. Most pathologically confirmed subsolid nodules were pulmonary adenocarcinomas [8,9,10], which were categorized into precursor glandular lesions (atypical adenomatous hyperplasia (AAH) and adenocarcinoma in situ (AIS)), minimally invasive adenocarcinomas (MIAs), and invasive adenocarcinomas (IAs) [11,12,13]. The precursor glandular lesions and MIAs were classified as non-invasive adenocarcinomas (non-IAs) because their 5-year recurrence-free survival (RFS) rate was 100% [14,15]. In comparison, the 5-year RFS rate of IAs ranged from 22.0% to 94.4% due to different differentiation grades [16,17]. 

Although pGGNs tend to be non-IAs and solid components within nodules correspond to invasive patterns in pathology, this association is not absolute [12,18]. In previous literature, a number of pGGNs were pathologically diagnosed as IAs, and the proportion ranged from 18.0% to 53.0% [19,20,21,22,23]. The current recommendations and guidelines maintain conservative attitudes and suggest annual screening with LDCT for pGGNs, as they tend to be stable or grow slowly during surveillance [18,24,25,26,27]. However, previous studies also found that over 50% of pGGNs progressed during the follow-up surveillance [28,29]. The invasiveness may be an indication that the pulmonary adenocarcinoma transitions from an indolent stage to a growth period. Hence, early differentiation of IAs from non-IAs in pGGNs is important for thoracic surgeons and radiologists when choosing surgical intervention or conservative surveillance.

Generally, radiologists evaluate the invasiveness of pGGNs by interpreting the morphological and quantitative characteristics using chest computed tomography (CT). As there is inherent subjectivity and inter-observer heterogeneity of the morphological features, most studies used the nodule size, usually the maximal or mean diameter, to distinguish IAs from non-IAs in pGGNs. However, the diagnostic performance of the nodule diameter was inconsistent. The area under the curve (AUC) ranged from 0.70 to 0.93, and the corresponding cut-off value ranged from 8.5 to 17.2 mm [20,30,31,32,33,34]. The heterogeneity of these results may be underlying the clinical characteristics of the sample, the acquisition parameters, and the measure of the maximal or mean diameter.

Therefore, this study aimed to assess the diagnostic performance of the nodule diameter for predicting invasiveness in pGGNs by performing a meta-analysis and to explore the potential heterogeneity. We also investigated whether the results of our meta-analysis could be validated in an independent cohort from a lung cancer screening.

## 2. Materials and Methods

### 2.1. Search Strategy and Selection Criteria

This meta-analysis was reported according to the preferred reporting items for systematic reviews and meta-analyses (PRISMA) guidelines [35]. We identified potentially eligible studies through electronic literature searches on PubMed, MEDLINE, Embase, and the Cochrane Library, from inception until December 2022. The following medical subject headings and keywords were used as search terms: (“computed tomography”) and (“adenocarcinoma”) and (“invasive *”) and (“ground-glass nodule *” or “ground glass nodule *” or “ground-glass opacity” or “ground glass opacity” or “non-solid nodule *” or “nonsolid nodule *” or “sub-solid nodule *” or “subsolid nodule *”). Further eligible studies were identified by screening the references in the retrieved original papers, review articles, and meta-analyses. The inclusion criteria were as follows: (1) using nodule maximal or mean diameter to differentiate IAs from non-IAs in pGGNs; (2) using histopathological examination as the gold standard of diagnosis; (3) reporting the sensitivity and specificity to calculate the true positive (TP), false positive (FP), false negative (FN), and true negative (TN). The exclusion criteria were as follows: (1) case reports or reviews; (2) duplicate publications or data; (3) studies including benign nodules; (4) studies unrelated to the nodule diameter; (5) insufficient data reporting; (6) differentiation between the AAH/AIS and MIA/IA; (7) studies including mGGNs.

### 2.2. Data Extraction

Two reviewers (J.L. and X.Y.) independently collected the following data from each included study: first author name, publication year and journal, the country of study, sample size, mean or median age, number of males, number of smokers, measure of the diameter, the cut-off value, and the acquisition parameters of the CT, including the slice thickness and reconstruction matrix. The following diagnostic performance measurements were calculated from the sensitivity and specificity: TP, FP, FN, and TN. Disagreement between the two reviewers was resolved by consulting a third reviewer (P.Z.).

### 2.3. Quality Assessment

The quality of the selected studies and the potential bias were assessed using the Quality Assessment of Diagnostic Accuracy Studies (QUADAS-2) tool [36]. This quality assessment procedure was independently performed by two reviewers (J.L. and X.Y.) and was checked by a third reviewer (P.Z.). Any disagreements were resolved through a discussion involving all the reviewers.

### 2.4. Meta-Analysis

The meta-analyses of the pooled sensitivity and specificity were performed using the MIDAS package in STATA (version 17.0), with a bivariate mixed-effect regression model. A summary receiver operating characteristic (SROC) plot was constructed to calculate the pooled AUC.

A meta-regression analysis was further conducted to explore the causes of heterogeneity using several potential covariates, including the percentage of males, percentage of smokers, measure of the diameter (maximal or mean), slice thickness (all < 1.5 mm or not), and reconstruction matrix (1024 × 1024 or 512 × 512). The patient selection of QUADA-2 was targeted as an additional covariate.

The heterogeneity of the included studies was assessed using a forest plot and the corresponding inconsistency index (I^2^). Moreover, I^2^ > 50% indicated a high degree of heterogeneity [37,38]. Publication bias was assessed with Egger’s funnel plot and with a regression test for funnel plot asymmetry [39].

### 2.5. Validation Using an Independent Sample

To verify the results from the meta-analysis, we conducted a validation study using an independent cohort from a lung cancer screening. The validation study was approved by the Ethics Committees at Sichuan Cancer Hospital, and individual consent for this retrospective study was waived. Our previous study, which included an eligible sample from March 2018 to December 2020, was also included in the meta-analysis [40], thus our validation cohort was enrolled from January 2021 to December 2022. Finally, a total of 210 pathologically confirmed pGGNs (82 IAs and 128 non-IAs) were consecutively enrolled from our institution to construct the validation cohort. 

All the patients underwent chest LDCT scans using a second-generation dual-source CT system (Somatom Definition Flash, Siemens Healthcare, Forchheim, Germany). The acquisition parameters of the LDCT were as follows: tube voltage, 100 kV; tube current, 10 to 30 mA; pitch, 1.0; collimation, 64 × 0.625 mm; rotation time, 0.33 s; field of view, 350 mm × 350 mm; iterative reconstruction algorithm (SAFIRE, strength level 5, Siemens Healthcare) with a soft reconstruction kernel (B); slice thickness, 0.5 mm; slice increment, 0.5 mm; reconstruction matrix, 512 × 512. All the patients had an LDCT scan within 1 month before surgical resection.

The uAI platform (United Imaging Healthcare, Shanghai, China), an artificial intelligence (AI) software based on deep learning methods [41,42], was used to automatically detect and segment pGGNs in three dimensions. The segmentation results were assessed by two radiologists (J.L. and H.Q.) in the lung window (window -500 HU, width 1500 HU). As all the segmentation results were satisfactory to both radiologists, no manual adjustments of the segmentation results were conducted to avoid inter- and intra-observer variability. Both the maximal diameter and mean diameter were recorded. To evaluate the consistency of the diameters produced by the AI software and the radiologist, 63 nodules (30% of 210) were randomly selected and measured by a third radiologist (P.Z.) who was blinded to the records of the AI software.

Statistical analysis was performed with MedCalc (version 18.2.1). The categorical variables were analyzed using Fisher’s test and the continuous variables were analyzed using the independent sample t-test. The agreement of the diameters produced by the AI software and the radiologist was evaluated using the intra-class correlation coefficient (ICC). To validate the diagnostic performance of the nodule diameter for differentiating IAs from non-IAs, we prespecified the cut-off value by calculating the average of the cut-off values from the included studies in the meta-analysis, weighting by the sample size. The comparisons of sensitivity and specificity were performed using the McNemar test [43].

## 3. Results

### 3.1. Characteristics of the Included Studies

The literature search and study selection included 16 studies in the meta-analysis (Figure 1). One study measured both the maximal diameter and mean diameter in the same sample [44], and another one used both the manual method and an automatic algorithm to measure the mean diameter in the same sample [45]. In these conditions, the measurement with a higher AUC was included in the pooled meta-analysis. In addition, one study conducted subgroup analysis with a duplicate sample and, thus, the subgroup with **a** larger sample was included [46]. The details of the study characteristics are presented in Table 1.

A total of 2564 pGGNs (761 IAs and 1803 non-IAs) were finally included in our meta-analysis. The TP, FP, FN, TN, and cut-off value from each report are presented in Table 2.

### 3.2. Quality Assessment

The QUADAS-2 results from the included studies are summarized in Table 3. Regarding patient selection, five studies had high risk of bias and applicability concerns as they did not include pGGNs of all sizes [20,31,49] or locations [23], or excluded pGGNs that were diagnosed with AAHs [44]. Regarding the index test, three studies did not report whether the readers were blinded to the results reference standard [30,31,46]. In addition, the cut-off values from all the studies were not prespecified, which might lead to overestimation of the diagnostic performance. However, few variations were found in the test technology, execution, or interpretation among these studies and, thus, their overall applicability was sufficient. As for the reference standard, five studies had an unclear risk of bias as they did not report the details of the histopathological assessment [22,31,32,34,48]. In regard to flow and timing, seven studies had an unclear risk of bias as they did not report the time interval between the CT scan and surgery [20,29,31,33,34,44,48].

### 3.3. Meta-Analysis of Diagnostic Performance

The sensitivity and specificity of the individual studies varied widely, ranging from 0.61 to 0.96 and 0.47 to 0.91. The pooled sensitivity and specificity were 0.82 (95% confidence interval (CI), 0.78–0.86) and 0.73 (95% CI, 0.67–0.78). The forest plots for all the included studies are shown in Figure 2. The AUC of the SROC was 0.85 (95% CI, 0.82–0.88) (Figure 3).

The results of the meta-regression analysis are shown in Table 4. The percentage of males, percentage of smokers, and slice thickness had no effect on the sensitivity or specificity (all *p* > 0.05). The mean diameter showed higher sensitivity and specificity compared with the maximal diameter (both *p* < 0.01). The reconstruction matrix of 1024 × 1024 showed higher specificity than that of 512 × 512 (*p* = 0.01), but no significant difference in the sensitivity was found between the two reconstruction matrices (*p* = 0.11). A high risk of patient selection was associated with significantly lower sensitivity but higher specificity than a low risk of patient selection (*p* = 0.04 and *p* < 0.01).

The I^2^ for sensitivity and specificity were 44.43% and 77.58%, indicating moderate to high heterogeneity among the included studies. The regression test for funnel plot asymmetry was insignificant (*p* = 0.54), suggesting a lack of publication bias.

### 3.4. Validation Using an Independent Sample

The characteristics of the pGGNs in the independent cohort are presented in Table 5. The age, maximal diameter, and mean diameter of the IA group were significantly higher than those of the non-IA group (all *p* < 0.01). No significant difference was found in regard to gender between the groups (*p* = 0.88). The ICCs of the maximal diameter and mean diameter between the AI software and the radiologist were 0.97 (0.96–0.98) and 0.98 (0.96–0.99), suggesting an excellent agreement.

As the meta-regression results showed that the measure of the diameter affected the diagnostic accuracy, we conducted validation tests using the maximal diameter and the mean diameter, respectively. The prespecified cut-off value was 13.2 mm for the maximal diameter and 10.4 mm for the mean diameter in the pGGNs. In the validation tests, the sensitivity was 0.72 (0.61–0.81) and the specificity was 0.86 (0.79–0.91) when using the maximal diameter. The sensitivity was 0.85 (0.78–0.91) and the specificity was 0.83 (0.75–0.89) when using the mean diameter (Table 6). The McNemar test further showed the mean diameter had higher sensitivity than the maximal diameter (*p* < 0.01), but no significant difference in the specificity was found between the two measures of the diameter (*p* = 0.13).

## 4. Discussion

This meta-analysis evaluated the diagnostic performance of the nodule diameter in predicting the invasiveness of pulmonary adenocarcinomas in pGGNs. We found that the pooled AUC, sensitivity, and specificity were 0.85, 0.82, and 0.73. Using the prespecified cut-off value of 13.2 mm for the maximal diameter and 10.4 mm for the mean diameter, our validation cohort of pGGNs showed that the sensitivity and specificity are 0.72 and 0.86 for the maximal diameter, and 0.85 and 0.83 for the mean diameter. Further comparisons showed the mean diameter had higher sensitivity than the maximal diameter. These results suggested that discriminating IAs from non-IAs in pGGNs was feasible using the nodule diameter, and the mean diameter with a cut-off value of 10.4 mm was recommended.

Radiologists usually assess the invasive probability of subsolid nodules using morphological and quantitative features via chest CT. Many morphological features were found to be related to the invasiveness of pGGNs. For example, lobulation was more frequently seen in IAs than Non-IAs [22,50], which was attributed to different rates of growth. The vacuole sign, with histological characteristics of collapse and dilated bronchioles, was highly suggestive of the invasiveness of pGGNs [40,50]. However, previous meta-analysis demonstrated that morphological features, such as the vacuole, speculation, lobulation, and pleural indentation, had inadequate diagnostic value for predicting invasiveness in subsolid nodules [51]. The AUCs, sensitivities, and specificities ranged from 0.60 to 0.67, 0.41 to 0.52, and 0.56 to 0.63. Besides, the morphological features had inter-observer heterogeneity, which was dependent on the subjectivity and experience of the radiologists. 

Compared with the morphological features, the quantitative features had better reproducibility with the application of computer-aided diagnosis. As one of the quantitative features, the mean CT value showed a sensitivity of 0.75 and a specificity of 0.81 in predicting invasiveness in subsolid nodules according to a recent meta-analysis, and the optimal cut-off value was −557 HU [52]. However, this meta-analysis included only six studies that contained both pGGNs and mGGNs, which might limit the use of the optimal cut-off value. In recent years, many studies have employed CT-derived radiomic features to differentiate IAs from non-IAs in subsolid nodules, and these radiomic models have shown excellent diagnostic performance [53,54,55,56,57]. The maximal AUC was 0.98 [54]. As radiomics requires additional software, the complexity of this approach limited its integration into clinical practice. Therefore, the nodule diameter, with the balance of objectivity and simplicity, may have diagnostic utility in predicting the invasiveness of pulmonary adenocarcinomas in pGGNs.

Regarding the Lung Imaging Reporting and Data System (Lung-RADS, v2022) and the Fleischner guideline, the diameter was a key feature for the management of pGGNs [18,58]. However, there were discrepancies in management strategies, including the cut-off value of the mean diameter and the interval of screening. Lung-RADS recommends that pGGNs with a mean diameter of < 30 mm should be annually screened. According to the Fleischner guideline, pGGNs with a mean diameter of < 6 mm require no routine screening, and those with a mean diameter of ≥6 mm should be screened at 6 to 12 months to confirm persistence and then be screened every 2 years. Our results indicated that the mean diameter of 10.4 mm might be the cut-off value between indolence and invasiveness, which could be a potential marker to determine the management strategy for pGGNs. In addition, biopsy was not recommended for pGGNs due to inadequate sampling and false-negative results [18]. 

According to the results of the meta-regression, the mean diameter showed higher sensitivity and specificity compared with the maximal diameter. In our validation test, the mean diameter showed higher sensitivity than the maximal diameter, which was similar to a previous study [44], and no significant difference was found in regard to the specificity. The current guidelines also recommend using the mean diameter to stratify the risk of the pulmonary nodules [18,25,27,58]. According to our results, the optimal cut-off value for the mean diameter was 10.4 mm for predicting invasiveness in pGGNs. However, as no previous studies conducted statistical comparisons between the diagnostic performance of the mean diameter and the maximal diameter in the same cohort, our results needed further validation using a large sample. The meta-regression results also demonstrated that a larger reconstruction matrix size was associated with higher specificity, which might result from a less partial volume effect, higher spatial resolution, and more accurate measurement of the nodule size [59]. However, this meta-regression result should be taken with caution, being driven by only four studies.

A high risk of patient selection was associated with lower sensitivity but higher specificity than a low risk of patient selection. Not enrolling all eligible pGGNs in the analysis had inherent bias and limited the diagnostic utility in the clinical workflow. Specifically, three of the five studies with a high risk of patient selection excluded small pGGNs (<8 or 10 mm) and resulted in a relatively high cut-off value for the maximal diameter, ranging from 14.0 to 16.4 mm [20,31,49]. The increase in the cut-off value might lead to an increase in missed diagnosis and a decrease in the FP rate for IAs. In addition, all included studies had a high risk of index test as the cut-off values were not prespecified. Therefore, the cut-off values were various, ranging from 8.5 to 17.2 mm for the maximal diameter and 9.8 to 10.8 mm for the mean diameter. In our validation test, we used the prespecified cut-off values derived from previous studies to avoid the risk of bias regarding the index test.

There were several limitations. First, one study with a relatively small sample (81 pGGNs) did not report the cut-off value for the maximal diameter [45], which might slightly affect the prespecified cut-off value used in our validation cohort. Second, the reconstruction matrix was fixed to 512 × 512 in the validation cohort. Further study to validate the effect of this acquisition parameter on diagnostic performance is required.

## 5. Conclusions

In conclusion, this meta-analysis showed that the nodule diameter had adequate diagnostic performance in differentiating IAs from non-IAs in pGGNs and could be replicated in a validation cohort. The mean diameter with a cut-off value of 10.4 mm was recommended.

## Figures and Tables

**Figure 1 diagnostics-14-00147-f001:**
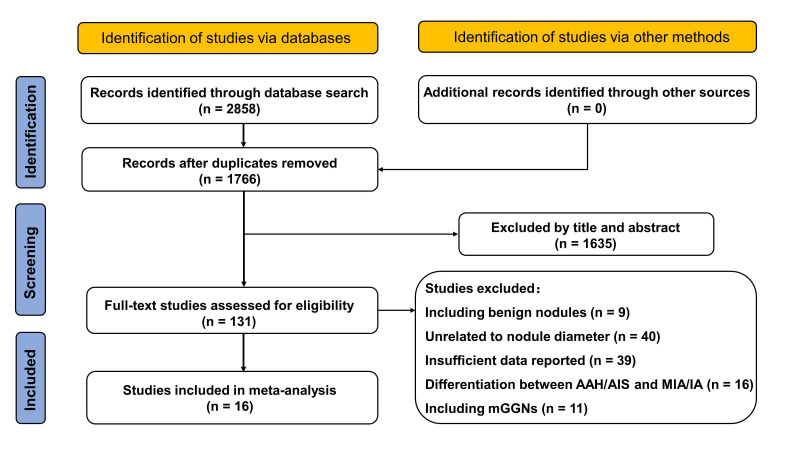
The flowchart of literature search and study selection. AAH: atypical adenomatous hyperplasia; AIS: adenocarcinoma in situ; MIA: minimally invasive adenocarcinoma; IA: invasive adenocarcinoma; mGGNs: mixed ground-glass nodules.

**Figure 2 diagnostics-14-00147-f002:**
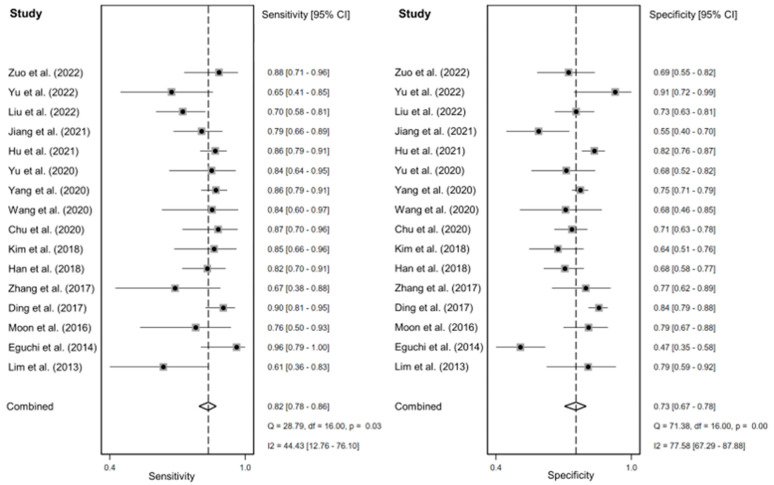
Forest plots on the sensitivity and specificity of the nodule size in predicting the invasiveness of pulmonary adenocarcinomas in pure ground-glass nodules. [20,22,23,29,30,31,32,33,34,40,44,45,46,47,48,49]. CI: confidence interval.

**Figure 3 diagnostics-14-00147-f003:**
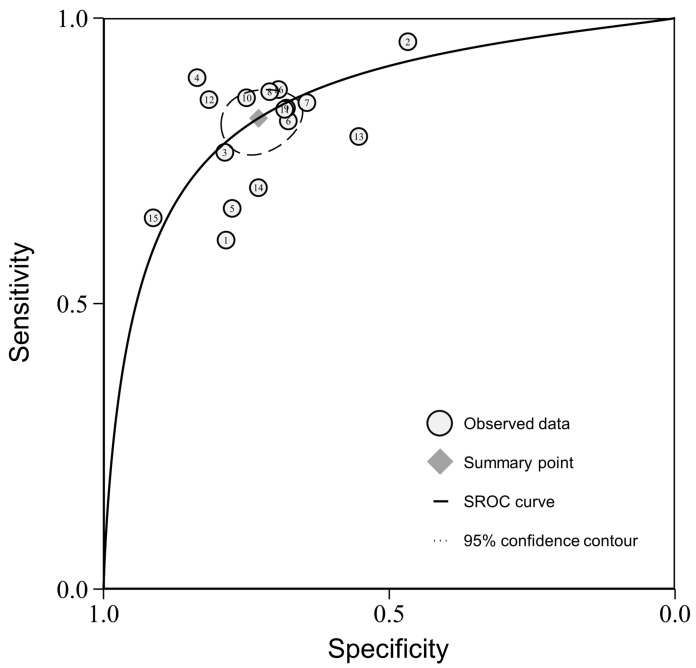
The summary receiver operating characteristic curve (SROC) plot on the diagnostic performance of the nodule size in predicting the invasiveness of pulmonary adenocarcinomas in pure ground-glass nodules.

**Table 1 diagnostics-14-00147-t001:** Characteristics of the studies included in the meta-analysis.

Study (Year)	Country	No. of Patients	No. of pGGNs		Age (Years)	No. of Males (%)	No. of Smokers (%)	Measure of Diameter	Slice Thickness (mm)	Matrix
			IA	non-IA						
Lim et al. (2013) [20]	Korea	46	18	28	61.4	26 (56.5)	14 (30.4)	maximal	0.75–2.5	NA
Eguchi et al. (2014) [29]	Japan	98	24	77	64.3	39 (38.6)	31 (30.7)	maximal	1.25	NA
Moon et al. (2016) [47]	Korea	83	17	66	58.4	31 (37.3)	19 (22.9)	maximal	NA	NA
Ding et al. (2017) [32]	China	NA	86	275	54.5	125 (34.6)	NA	maximal	1.0	NA
Zhang et al. (2017) # [31]	China	53	15	40	59.0 *	13 (24.5)	0 (0)	maximal	1.25	NA
Han et al. (2018) # [34]	China	154	61	102	55.2	52 (33.8)	NA	maximal	1.25	NA
Kim et al. (2018) [46]	Korea	86	27	59	NA	41 (47.7)	NA	mean	0.625–1.25	NA
Chu et al. (2020) [22]	China	161	31	141	53.4	48 (27.9)	29 (16.9)	mean	0.625	NA
Wang et al. (2020) [30]	China	44	19	25	NA	NA	NA	maximal	0.9	1024 × 1024
Yang et al. (2020) [44]	China	641	136	523	NA	200 (30.3)	309 (46.9)	mean	NA	1024 × 1024
Yu et al. (2020) # [48]	China	62	25	41	55.4	19 (30.6)	4 (6.5)	maximal	1.25	NA
Hu et al. (2021) [33]	China	309	133	211	53.4	98 (28.5)	NA	mean	1.0	NA
Jiang et al. (2021) [23]	China	100	53	47	60.5 *	29 (29.0)	8 (8.0)	maximal	1.0–1.5	512 × 512
Liu et al. (2022) [40]	China	160	64	96	51.4	54(33.8)	NA	mean	0.625	512 × 512
Yu et al. (2022) # [49]	China	42	20	23	56.4	8 (19.1)	NA	maximal	1.0	NA
Zuo et al. (2023) # [45]	China	68	32	49	52.6	18(26.5)	NA	maximal	0.625–1.25	NA

# The median or mean age, percentage of males, and percentage of smokers in the studies are calculated according to the number of patients, and those of the others are calculated according to the number of nodules. * The ages are shown as the median, and the others as the mean. pGGNs: pure ground-glass nodules; IA: invasive adenocarcinoma; non-IA: non-invasive adenocarcinoma; NA: not available.

**Table 2 diagnostics-14-00147-t002:** TP, FP, FN, TN, and cut-off value from the reports included in the meta-analysis.

Study	TP	FP	FN	TN	Cut-off (mm)
Lim et al. (2013) [20]	11	6	7	22	16.4
Eguchi et al. (2014) [29]	23	41	1	36	11.0
Moon et al. (2016) [47]	13	14	4	52	15.0
Ding et al. (2017) [32]	77	45	9	230	12.0
Zhang et al. (2017) [31]	10	9	5	31	16.1
Han et al. (2018) [34]	50	33	11	69	17.2
Kim et al. (2018) [46]	23	21	4	38	10.4
Chu et al. (2020) [22]	27	41	4	100	10.5
Wang et al. (2020) [30]	16	8	3	17	8.5
Yang et al. (2020) [44]	117	131	19	392	10.1
Yu et al. (2020) [48]	21	13	4	28	9.4
Hu et al. (2021) [33]	114	39	19	172	9.8
Jiang et al. (2021) [23]	42	21	11	26	13.9
Liu et al. (2022) [40]	45	26	19	70	10.0
Yu et al. (2022) [49]	13	2	7	21	14.0
Zuo et al. (2023) [45]	28	15	4	34	NA

TP: true positive; FP: false positive; TN: true negative; FN: false negative.

**Table 3 diagnostics-14-00147-t003:** Quality assessment of the studies included in the meta-analysis.

Study	Risk of Bias				Applicability Concerns		
	Patient Selection	Index Test	Reference Standard	Flow and Timing	Patient Selection	Index Test	Reference Standard
Lim et al. (2013) [20]	-	-	+	?	-	+	+
Eguchi et al. (2014) [29]	+	-	+	?	+	+	+
Moon et al. (2016) [47]	+	-	+	+	+	+	+
Ding et al. (2017) [32]	+	-	?	+	+	+	?
Zhang et al. (2017) [31]	-	-	?	?	-	+	?
Han et al. (2018) [34]	+	-	?	?	+	+	?
Kim et al. (2018) [46]	+	-	?	+	+	+	+
Chu et al. (2020) [22]	+	-	?	+	+	+	?
Wang et al. (2020) [30]	+	-	?	+	+	+	+
Yang et al. (2020) [44]	-	-	+	?	-	+	+
Yu et al. (2020) [48]	+	-	?	?	+	+	?
Hu et al. (2021) [33]	+	-	+	?	+	+	+
Jiang et al. (2021) [23]	-	-	+	+	-	+	+
Liu et al. (2022) [40]	+	-	+	+	+	+	+
Yu et al. (2022) [49]	-	-	+	+	-	+	+
Zuo et al. (2023) [45]	+	-	+	+	+	+	+

Index: + low risk, - high risk, ? unclear risk.

**Table 4 diagnostics-14-00147-t004:** Meta-regression analysis of related covariates.

Covariates	No. of Reports		Sensitivity (95% CI)	*p*	Specificity (95% CI)	*p*
Percentage of males	16		0.73 (0.06–0.99)	0.77	0.38 (0.02–0.95)	0.38
Percentage of smokers	16		0.96 (0.60–1.00)	0.22	0.76 (0.22–0.97)	0.81
Measure of diameter	11	Maximal diameter	0.82 (0.76–0.87)	<0.01	0.72 (0.66–0.79)	<0.01
	5	Mean diameter	0.84 (0.77–0.90)		0.74 (0.66–0.82)	
Slice thickness	12	All < 1.5 mm	0.84 (0.79–0.88)	0.58	0.73 (0.67–0.79)	0.58
	2	Not all < 1.5 mm	0.73 (0.57–0.88)		0.67 (0.49–0.85)	
Reconstruction matrix	2	1024 × 1024	0.86 (0.80–0.91)	0.11	0.75 (0.71–0.78)	0.01
	2	512 × 512	0.74 (0.66–0.82)		0.67 (0.59–0.75`)	
Patient selection	11	Low risk	0.85 (0.81–0.89)	0.04	0.72 (0.66–0.78)	<0.01
	5	High risk	0.76 (0.67–0.85)		0.76 (0.67–0.85)	

CI: confidence interval.

**Table 5 diagnostics-14-00147-t005:** Characteristics of pGGNs in the independent sample.

Characteristics	IA (n = 82)	Non-IA (n = 128)	*p*
Gender (male/female)			0.88
Female	53	84	
Male	29	44	
Age	59.6 ± 10.5	49.2 ± 11.8	<0.01
Maximal diameter	16.7 ± 5.6	9.6 ± 3.4	<0.01
Mean diameter	14.8 ± 4.7	8.6 ± 2.9	<0.01

pGGNs: pure ground-glass nodules; IA: invasive adenocarcinoma; non-IA: non-invasive adenocarcinoma.

**Table 6 diagnostics-14-00147-t006:** Diagnostic performance of the nodule size for predicting the invasiveness of pulmonary adenocarcinomas in pGGNs in the independent sample.

Measures	Sensitivity (95% CI)	Specificity (95% CI)	Cut-off
Maximal diameter	0.72 (0.61–0.81)	0.86 (0.79–0.91)	> 13.2 mm
Mean diameter	0.85 (0.78–0.91)	0.83 (0.75–0.89)	> 10.4 mm

pGGNs: pure ground-glass nodules; CI: confidence interval.

## Data Availability

The data presented in the validation part of this study are available on request from the corresponding author. The data are not publicly available due to privacy.

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
