# Peer review of "Predicting the Invasiveness of Pulmonary Adenocarcinomas in Pure Ground-Glass Nodules Using the Nodule Diameter: A Systematic Review, Meta-Analysis, and Validation in an Independent Cohort"

_diagnostics, 2024, doi:10.3390/diagnostics14020147_

Round 1

Reviewer 1 Report

Comments and Suggestions for Authors

The systematic review addresses a key issue in lung cancer diagnosis by demonstrating that measuring nodule size can differentiate between invasive and non-invasive lung cancers, with a suggested cut-off of 10.4 mm. This has the potential to significantly improve the accuracy of lung cancer screening and diagnosis.

Author Response

We appreciate the reviewer’s encouraging comments.

Reviewer 2 Report

Comments and Suggestions for Authors

A valuable, well designed and well written study.

Author Response

(The authors gave the same response as above.)

Reviewer 3 Report

Comments and Suggestions for Authors

A review of “Predicting Invasiveness of Pulmonary Adenocarcinomas in Pure Ground-Glass Nodules by Nodule Diameter: A Meta-Analysis and Validation in an Independent Cohort” paper by Jieke Liu, Xi Yang, Yong Li, Hao Xu, Changjiu He, Peng Zhou and Haomiao Qing

The main question of the paper “Predicting Invasiveness of Pulmonary Adenocarcinomas in Pure Ground-Glass Nodules by Nodule Diameter: A Meta-Analysis and Validation in an Independent Cohort” by M Jieke Liu, Xi Yang, Yong Li, Hao Xu, Changjiu He, Peng Zhou and Haomiao Qing is a meta-analysis to evaluate the diagnostic performance of pulmonary nodule diameter for predicting invasiveness in pure ground glass nodules (pGGNs) found in lung computed tomography (CT) scans and subsequent validation of the cut-off value of diameter in an independent cohort.

The importance of this topic is due to the need in early differentiating invasive and non-invasive adenocarcinomas growing from pGGNs. The diagnostic performance of nodule diameter that is currently used for this purpose is inconsistent.

The body of this meta-analysis is appropriate to its goals. The authors used literature search in large electronic databases with two independent reviewers on the basis of systematic reviews and meta-analyses (PRISMA) guideline. The results from the meta-analysis were subsequently validated using an independent cohort from lung cancer screening.

The meta-analysis included 16 studies of sufficient quality and 2564 pGGNs in total. The validation cohort included 210 patients with invasive and non-invasive adenocarcinomas.

The conclusions made by the authors are consistent with the aim of the meta-analysis and results obtained.

The references are appropriate to the topic and the content of the meta-analysis.

The meta-analysis includes six tables and three figures that are consistent with the main task of the meta-analysis. The data are shown in a clear manner.

Therefore, the review could be published without any remarks.

Author Response

(The authors gave the same response as above.)

Reviewer 4 Report

Comments and Suggestions for Authors

1. Article title: Predicting Invasiveness of Pulmonary Adenocarcinomas in Pure Ground-Glass Nodules by Nodule Diameter: A Meta-Analysis and Validation in An Independent Cohort
2. The study is highly relevant to the fields of oncology and chest imaging
3. The main strength is excellent methodology, allowing for result reproducibility. Primary weakness are related to (1) discussion lacking in examples of similar CT features (i.e., signs predicting histological characteristics) implemented into clinical pathways. Would mean GGN diameter really alter patient management? Is there such precedent when CT alleviates the need for surveillance and/or biopsy? And (2) issues with measurement reproducibility (between various artificial intelligence algorithms as well as human readers). Margin of error is unclear, limiting potential applicability. Consider providing clarifications on these two point in order to further strengthen the study.
4. The tables and figures are informative.

Comments on the Quality of English Language

Consider performing an additional language check using LanguageTool or similar application in order to eliminate typos.

Author Response

We would like to thank the reviewer for the encouraging, helpful and considered comments. We feel that responding to these comments has made the paper considerably stronger. Changes have been marked in blue and bold in the revised manuscript. Summary of our detailed responses are given below.

Comments 1:

Primary weakness are related to (1) discussion lacking in examples of similar CT features (i.e., signs predicting histological characteristics) implemented into clinical pathways. Would mean GGN diameter really alter patient management? Is there such precedent when CT alleviates the need for surveillance and/or biopsy?

Response 1:

Firstly, we added the discussion of representative morphological features that were related to the invasiveness as follows:

Many morphological features were found to be related to the invasiveness of pGGNs. For example, lobulation was more frequently seen in IAs than Non-IAs [1,2], which was attributed to different rates of growth. Vacuole sign, with histological characteristics of collapse and dilated bronchioles, was highly suggestive of the invasiveness of pGGNs [2,3] (Revised manuscript/Lines 248-252).

Secondly, the importance of diameter in determining management strategy for pGGNs was also been added in the discussion part as follows:

Regarding the Lung-Imaging Reporting and Data System (Lung-RADS, v2022) and the Fleischner guideline, diameter was a key feature for the management of pGGNs [4,5]. However, there were discrepancies in management strategies, including the cut-off value of mean diameter and the interval of screening. The Lung-RADS recommended that pGGNs with a mean diameter of < 30 mm should be annually screened. According to the Fleischner guideline, pGGNs with a mean diameter of < 6 mm required no routine screening, and those with a mean diameter of ≥ 6 mm should be screened at 6 to 12 months to confirm persistence and then be screened every 2 years. Our results indicated the mean diameter of 10.4 mm might be the cut-off value between indolence and invasiveness, which could be a potential marker to determine management strategy for pGGNs. Besides, biopsy was not recommended for pGGNs due to inadequate sampling and false-negative results [4] (Revised manuscript/Lines 272-283).

Comments 2:

And (2) issues with measurement reproducibility (between various artificial intelligence algorithms as well as human readers). Margin of error is unclear, limiting potential applicability.

Response 2:

To address the measurement reproducibility issue and to clarity the margin of error, we performed the consistency analysis between AI software and the radiologist. The added contents were as follows:

To evaluate the consistency of the diameters between AI software and the radiologist, 63 nodules (30% of 210) were randomly selected and measured by a third radiologist (P.Z.) who was blinded to the records of AI software (Revised manuscript/Lines 141-144). The agreement of diameters between AI software and the radiologist was evaluated by the intra-class correlation coefficient (ICC) (Revised manuscript/Lines 147-148). The ICCs of maximal diameter and mean diameter between AI software and the radiologist were 0.97 (0.96 - 0.98) and 0.98 (0.96 - 0.99), suggesting an excellent agreement (Revised manuscript/Lines 219-221).

Comments 3:

Comments on the Quality of English Language: Consider performing an additional language check using LanguageTool or similar application in order to eliminate typos.

Response 3:

The language check has been conducted using LanguageTool, and the typos have been revised.

References

  1. Chu, Z.G.; Li, W.J.; Fu, B.J.; Lv, F.J. CT Characteristics for Predicting Invasiveness in Pulmonary Pure Ground-Glass Nodules. AJR Am J Roentgenol 2020, 215, 351-358, doi:10.2214/ajr.19.22381.
  2. Hsu, W.C.; Huang, P.C.; Pan, K.T.; Chuang, W.Y.; Wu, C.Y.; Wong, H.F.; Yang, C.T.; Wan, Y.L. Predictors of Invasive Adenocarcinomas among Pure Ground-Glass Nodules Less Than 2 cm in Diameter. Cancers (Basel) 2021, 13, doi:10.3390/cancers13163945.
  3. Liu, J.; Yang, X.; Li, Y.; Xu, H.; He, C.; Qing, H.; Ren, J.; Zhou, P. Development and validation of qualitative and quantitative models to predict invasiveness of lung adenocarcinomas manifesting as pure ground-glass nodules based on low-dose computed tomography during lung cancer screening. Quant Imaging Med Surg 2022, 12, 2917-2931, doi:10.21037/qims-21-912.
  4. MacMahon, H.; Naidich, D.P.; Goo, J.M.; Lee, K.S.; Leung, A.N.C.; Mayo, J.R.; Mehta, A.C.; Ohno, Y.; Powell, C.A.; Prokop, M.; et al. Guidelines for Management of Incidental Pulmonary Nodules Detected on CT Images: From the Fleischner Society 2017. Radiology 2017, 284, 228-243, doi:10.1148/radiol.2017161659.
  5. American College of Radiology. Lung CT Screening Reporting and Data System (Lung-RADS, v2022). Available online: https://www.acr.org/Clinical-Resources/Reporting-and-Data-Systems/Lung-Rads.